# Glucose intake hampers PKA-regulated HSP90 chaperone activity

Yu-Chen Chen[1], Pei-Heng Jiang[1], Hsuan-Ming Chen[1], Chang-Han Chen[1], Yi-Ting Wang[2], Yu-Ju Chen[2], Chia-Jung Yu[3,4], Shu-Chun Teng[1,5]*

[1]Department of Microbiology, College of Medicine, National Taiwan University, Taipei, Taiwan; [2]Institute of Chemistry, Academia Sinica, Taipei, Taiwan; [3]Department of Cell and Molecular Biology, College of Medicine, Chang Gung University, Tao-Yuan, Taiwan; [4]Department of Thoracic Medicine, Chang Gung Memorial Hospital, Tao-Yuan, Taiwan; [5]Center of Precision Medicine, National Taiwan University, Taipei, Taiwan

**Abstract** Aging is an intricate phenomenon associated with the gradual loss of physiological functions, and both nutrient sensing and proteostasis control lifespan. Although multiple approaches have facilitated the identification of candidate genes that govern longevity, the molecular mechanisms that link aging pathways are still elusive. Here, we conducted a quantitative mass spectrometry screen and identified all phosphorylation/dephosphorylation sites on yeast proteins that significantly responded to calorie restriction, a well-established approach to extend lifespan. Functional screening of 135 potential regulators uncovered that Ids2 is activated by PP2C under CR and inactivated by PKA under glucose intake. *ids2Δ* or *ids2* phosphomimetic cells displayed heat sensitivity and lifespan shortening. Ids2 serves as a co-chaperone to form a complex with Hsc82 or the redundant Hsp82, and phosphorylation impedes its association with chaperone HSP90. Thus, PP2C and PKA may orchestrate glucose sensing and protein folding to enable cells to maintain protein quality for sustained longevity.

DOI: https://doi.org/10.7554/eLife.39925.001

*For correspondence:
shuchunteng@ntu.edu.tw

Competing interests: The authors declare that no competing interests exist.

## Introduction

Aging is a complex process in which cells gradually lose their ability to execute regular functions and organs show increased susceptibility to disease (*Harman, 1981*). The causes of aging have been attributed to nine hallmarks (*López-Otín et al., 2013*). However, crosstalk between these hallmarks is rare. Calorie restriction (CR) is the most effective intervention known to extend lifespan (*McCay et al., 1989*), improve organism function, and retard cell senescence in a variety of species from yeast to mammals (*McCay et al., 1989*; *de Cabo et al., 2015*; *de Cabo et al., 2014*; *Lin et al., 2002*). CR delays the onset and/or reduces the incidence of many age-related diseases, including cancer, diabetes, and cardiovascular disorders (*Mattson and Wan, 2005*; *Roth et al., 2001*).

Budding yeast has been perceived as an advantageous model to conveniently examine and isolate new components in aging-related pathways. In yeast, CR can be modeled by reducing the glucose concentration of the media from 2% as the *ad libitum* concentration to 0.5% (or lower), resulting in a 30–40% increase in lifespan (*Lin et al., 2002*; *Lin et al., 2000*). Two different lifespan paradigms are established in querying the lifespan of yeast cells: the replicative lifespan (*Tesch et al., 1978*) and chronological lifespan (CLS). RLS denotes the number of daughter cells that a single mother cell can generate before senescence, representing the division potential of the mother cell (*Mortimer and Johnston, 1959*), whereas CLS refers to the length of time for which yeast cells remain viable in a non-dividing state (*Longo and Fabrizio, 2012*).

CR exerts anti-aging effects by regulating metabolism (*Kapahi et al., 2004*) and enhancing stress resistance (*Fabrizio et al., 2001*). These processes down-regulate the amino acid-sensing mTOR and the glucose-sensing PKA signaling pathways (*Wei et al., 2008*). Attenuation of Tor1, Sch9, and PKA kinases promotes activation of Rim15 kinase-mediated transcription factors Msn2/4 and Gis1 to increase pathways in glycogen accumulation, antioxidant enzyme formation, heat shock protein expression, and autophagy (*Fontana et al., 2010*). Moreover, CR-mediated Tor1 repression enhances Snf1 kinase (the mammalian AMP kinase homologue in yeast) (*Orlova et al., 2006*) to extend CLS by promoting respiration, acetyl-CoA levels, and autophagy (*Lin et al., 2003*; *Wierman et al., 2017*; *Wang et al., 2001*; *Wright and Poyton, 1990*). Although these signaling pathways regulated by CR-modulated phosphorylation have been studied thoroughly, we speculate that there are other unknown regulators remaining to be explored. Here, we developed a screening procedure to discover additional pathways regulated under CR.

## Results

### A phosphoproteomic screening method identified Ids2 dephosphorylation under CR

To explore anti-aging mechanisms, we performed mass spectrometry-based quantitative phosphoproteomic profiling to globally define the phosphorylation and dephosphorylation sites of regulated proteins under CR. The triplicate phosphoproteome maps generated from each calorie-restricted or normal glucose-treated sample were annotated with the phosphopeptides of unambiguously identified phosphorylated amino acid sequences (*Figure 1—figure supplement 1*). Over 2672 unique phosphopeptides on 949 proteins were identified ($p < 0.05$, *Figure 1*–source file 1). The rightward shift in the 2% glucose/0.5% glucose ratio of phosphopeptides in *Figure 1—figure supplement 1B* indicated that the abundance of phosphopeptides decreased under 0.5% glucose, which is in agreement with that many kinases, such as Tor, Sch9, and PKA, were downregulated under CR (*Wei et al., 2008*). Among these phosphopeptides, 318 proteins (508 phosphopeptides) showed a 2-fold increase under 0.5% glucose, meaning that these peptides were phosphorylated under a glucose-limited CR environment; 427 proteins (825 phosphopeptides) showed a 2-fold decrease under 0.5% glucose, indicating that these peptides were phosphorylated under a glucose-enriched environment; and 113 proteins contained both increased and decreased phosphopeptides on the same protein under 0.5% glucose.

To discover additional pathways regulated by phosphorylation under CR, we designed an in silico and functional screening procedure (*Figure 1A*). *Harman, 1981* total of 632 proteins quantified with 2-fold differential phosphorylation to Gene Ontology (GO) analysis (*Mi et al., 2013*). Among them, 334 phosphoproteins were annotated to participate in biological processes including glycolysis ($p < 0.01$), cytokinesis ($p < 0.01$), signal transduction ($p < 1.0 \times 10^{-3}$), cell communication ($p < 1.0 \times 10^{-3}$), transcription ($p < 1.0 \times 10^{-5}$), and stress response ($p < 1.0 \times 10^{-4}$) (*Figure 1—figure supplement 1C*). Many instances of phosphorylation regulation in our database coincided with the existing CR/nutrient-sensing pathways (*Dilova et al., 2007*), such as PKA, Tor, and Snf1 (*Wei et al., 2008*) (*Figure 1—figure supplement 1D*). In previous studies, CR has been shown to extend lifespan through an increase in respiration (*Lin et al., 2002*), augmentation of cellular protection (*Wei et al., 2008*), and induction of autophagy (*Morselli et al., 2010*). Thus, we challenged the deletion strains of 135 candidates in the top categories of the GO analysis with various stresses, including heat shock, oxidative stress, DNA replicative stress, and/or nonfermentable carbon growth. We also verified the efficiency of autophagy by a GFP-Atg8 cleavage assay (*Cheong et al., 2005*). Phenotypic analyses exhibited that 53 deletion strains showed defective phenotypes (*Figure 1—source data 2* and *Figure 1—figure supplement 2*). Phosphomimetic and/or dephosphomimetic mutations of these 53 candidates were generated to verify the functions observed in the deletion strains, and only the *ids2-S148D* phosphomimetic strain displayed a growth defect in heat shock and glycerol (*Figure 1B*). According to our mass spectrometry data (*Figure 2—figure supplement 1A*), S148 on Ids2 was identified (Ids2 RRS$\underline{S}^{148}$IQDVQWIR) to be dephosphorylated under CR. The growth defect under CR was observed in *ids2Δ* and *ids2-S148D* cells (*Figure 1C*). Moreover, the *ids2-S148D* phosphomimetic strains showed shortened CLS (*Figure 1D*). Ids2 is involved in the modulation of Ime2 during meiosis (*Sia and Mitchell, 1995*); however, it is a functionally unknown protein in the mitotic

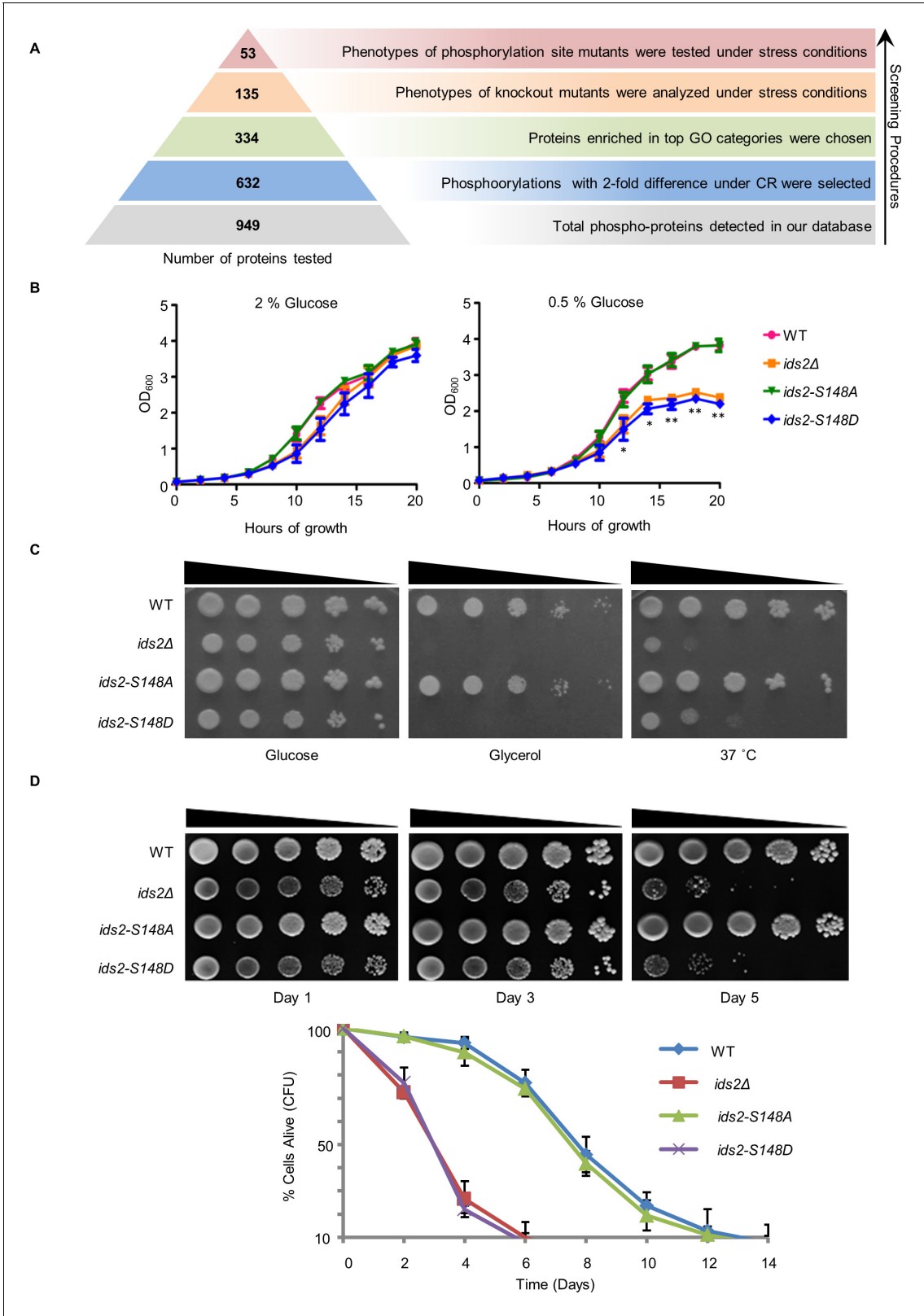

**Figure 1.** Ids2-S148 phosphorylation influences stress tolerance, growth, and chronological lifespan. (A) A schematic diagram highlights the steps in candidate screening. (B) Growth curves at 30°C were monitored in triplicate and represented as the mean ±S.E. (standard errors). (C) Tenfold serially diluted cells were grown under heat shock (37°C) in 2% glucose or at 30°C in 3% glycerol. WT and S148D mutants were compared using Student's

*Figure 1 continued on next page*

*Figure 1 continued*

t-test. *, p < 0.05. **, p < 0.01. (D) For the chronological lifespan assay, CFU viability was determined. 10-fold serial dilutions were spotted on YEPD plates. Quantitative CLS assay was assessed in triplicate by colony-forming capacity on YEPD plates.

DOI: https://doi.org/10.7554/eLife.39925.002

The following source data and figure supplements are available for figure 1:

**Source data 1.** The list of total phosphopeptides influenced by calorie restriction.

DOI: https://doi.org/10.7554/eLife.39925.005

**Source data 2.** Genes and mutation sites for functional screening.

DOI: https://doi.org/10.7554/eLife.39925.006

**Figure supplement 1.** The strategy used for large-scale identification of CR-modulated pathways.

DOI: https://doi.org/10.7554/eLife.39925.003

**Figure supplement 2.** Most phosphorylation site mutants displayed wild-type phenotypes under stress conditions.

DOI: https://doi.org/10.7554/eLife.39925.004

cell cycle. We, therefore, speculated that Ids2-S148 phosphorylation may play an important role in stress response and lifespan control.

## PKA phosphorylates Ids2 upon glucose intake, and PP2C dephosphorylates Ids2 under CR

Because the protein level of Ids2 did not change under CR (*Figure 2—figure supplement 1B*), we hypothesized that the phosphorylation status of Ids2 may modulate its function. To verify the CR-mediated Ids2 phosphorylation, an antibody against S148 phosphorylation was generated (*Figure 2—figure supplement 1C*). Western blot analysis showed that Ids2-S148 phosphorylation decreased under CR (*Figure 2A* and *Figure 2—figure supplement 1D*) but not under nitrogen starvation or TOR pathway inhibition (*Figure 2B* and *Figure 2—figure supplement 1D*). Based on the PhosphoGRID and pkaPS databases (*Neuberger et al., 2007*), Ids2 RRSS$^{148}$IQD may be phosphorylated by PKA (score = 1.57). Deletion of one PKA catalytic subunit, *TPK3*, reduced the Ids2 phosphorylation significantly (*Figure 2C*), suggesting that Tpk3 may phosphorylate S148. Deletion of *RAS2* or *GPA2*, the two PKA upstream regulators did not affect Ids2 phosphorylation (*Figure 2—figure supplement 1E*), implying that they act in a redundant manner. An in vitro kinase assay demonstrated that GST-Ids2 was phosphorylated by wild-type PKA kinase from the extract of the KT1115 strain (*Figure 2D*) or by the bovine heart PKA catalytic subunit C (*Figure 2—figure supplement 1F*) but not by the extract of the PKA kinase-dead strain (*Figure 2D*). These results indicated that PKA may directly phosphorylate Ids2-S148.

To determine the phosphatase(s) dephosphorylating Ids2-S148, we examined the phosphorylation levels in the mutants of major phosphatases. The Ids2 phosphorylation level increased in *ptc2Δ ptc3Δ* cells compared with that in the wild-type (*Figure 2E*). An in vitro phosphatase assay showed that recombinant Ptc2 removed the S148 phosphorylation (*Figure 2F*), indicating that PP2C may directly dephosphorylate Ids2. Moreover, a spotting assay showed that the *ids2-S148A* mutation could rescue the growth defect in the *ptc2Δ*, *ptc3Δ* and *ptc2Δ ptc3Δ* mutants (*Figure 2G*), suggesting that the PP2C phosphatases (*Sharmin et al., 2014*) may dephosphorylate Ids2.

## Ids2 interacts with chaperone HSP90 in a phosphorylation-modulated manner

To understand the molecular mechanism that Ids2 participates in under CR, we searched for its interaction partners via tandem affinity purification (TAP) and mass spectrometry analysis (*Liu et al., 2015*). The bead-bound proteins were separated (*Figure 3* and *Figure 3—figure supplement 1*), and spectrometry analysis revealed two closely related proteins, Hsc82 and Hsp82 (*Schopf et al., 2017*), in the yeast HSP90 family (*Figure 3*–source file 1). Expression of Hsc82 is tenfold higher than that of Hsp82 under normal conditions, suggesting that Hsc82 plays a more prominent role (*Schopf et al., 2017*). The co-immunoprecipitation assay showed that Hsc82-HA$_3$ co-precipitated with Ids2-Myc$_{13}$ (*Figure 3B*). A yeast two-hybrid analysis revealed that amino acids 92–256 of Ids2 interact with amino acids 272–579 of Hsc82 (*Figure 3—figure supplement 2*). Immunoprecipitation of Ids2-S148D with Hsc82 was significantly decreased, while the Ids2-S148A mutation increases the interaction with Hsc82 (*Figure 3C*). In addition, the phosphomimetic S148D mutation of purified

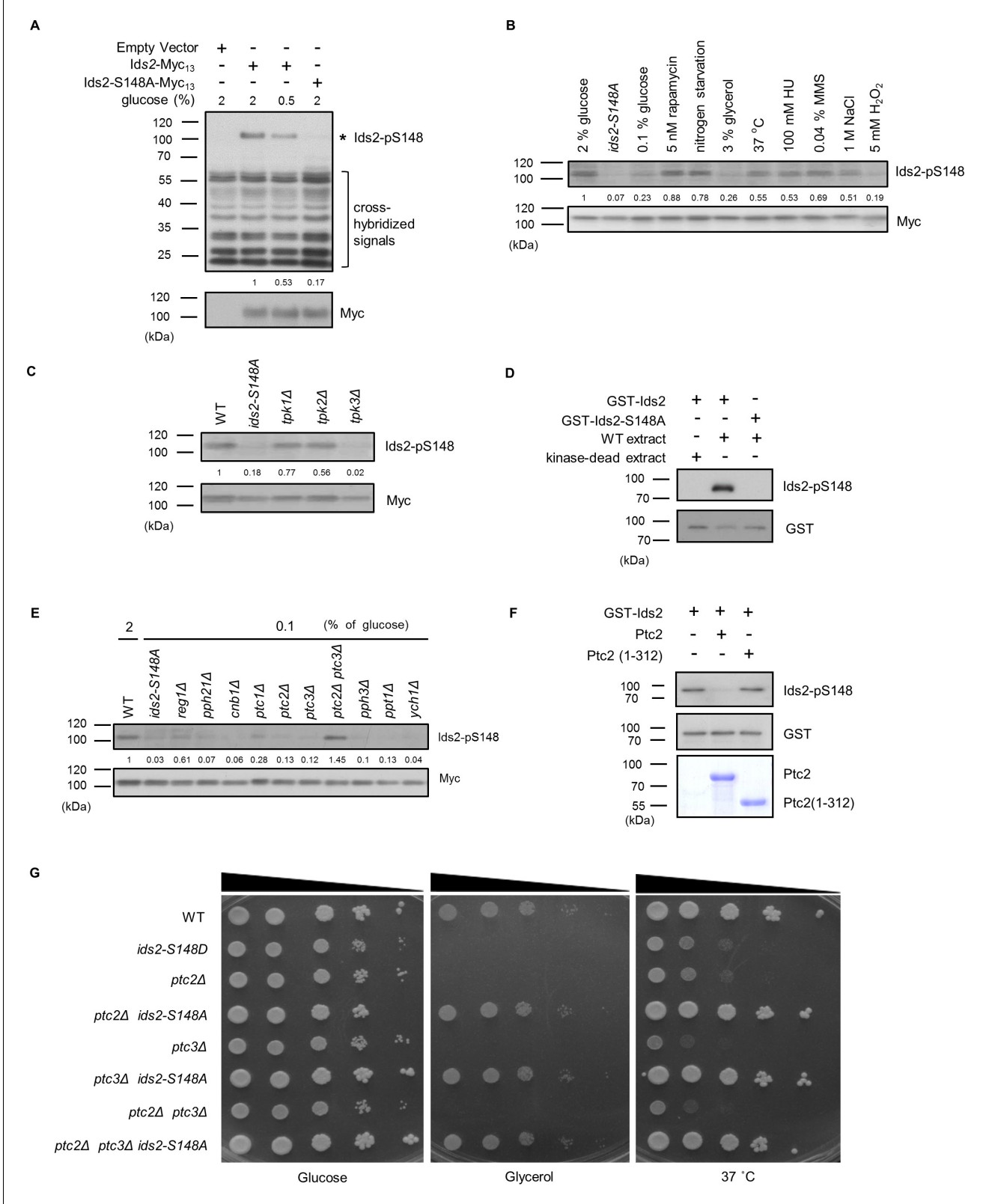

**Figure 2.** PKA and PP2C regulate Ids2-S148 phosphorylation.  (A–C, E) Strains were transformed with pRS426-Ids2-Myc$_{13}$, and lysates were examined by Western blot analysis with the indicated antibodies. The numbers below are the means of the intensity ratios of Ids2-p148/Myc compared with that of WT treated with 2% glucose. (B) Overnight cells were refreshed in medium with different treatments for 3 hr. (D,F) In vitro kinase and phosphatase

*Figure 2 continued on next page*

*Figure 2 continued*
assays were conducted as described in the Methods section. (G) Cells were spotted in 10-fold dilutions on YEPD or YEPG plates and grown at 30 or 37°C.
DOI: https://doi.org/10.7554/eLife.39925.007
The following figure supplement is available for figure 2:

**Figure supplement 1.** Characterization of the phosphospecific antibodies against Ids2 S148.
DOI: https://doi.org/10.7554/eLife.39925.008

recombinant Ids2 impaired the Ids2-Hsc82 interaction in the affinity pull-down assay (*Figure 3D*), suggesting that the phosphorylation of S148, which is located within the interacting motif (92-256), may inhibit the Ids2-Hsc82 interaction. To examine the interaction between Ids2 and Hsc82 under CR, an overnight culture was refreshed in medium with 2% or 0.5% glucose. As expected, CR significantly increased the Ids2-Hsc82 interaction (*Figure 3E*).

## Ids2 acts as a co-chaperone in the HSP90-mediated protein-folding process

HSP90 is an essential eukaryotic chaperone with a well-established role in folding and maintenance of proteins that are associated with the cell signaling response, such as kinases and hormone receptors (*Schopf et al., 2017*). However, the active participation of co-chaperone proteins at various stages of the chaperone folding cycle is crucial for the completion of the folding process (*Li et al., 2012*). Due to the physical interaction between Ids2 and Hsc82, we speculated that Ids2 may function as either a substrate or a regulator of HSP90. Deletion of *HSC82* did not change the stability of Ids2, suggesting that Ids2 may not be a substrate of Hsc82 (*Figure 4—figure supplement 1*). Many co-chaperones can regulate HSP90 ATPase, an activity essential for its chaperoning action (*Richter et al., 2004*; *Wolmarans et al., 2016*). To test whether Ids2 could modulate the ATPase activity of Hsc82, we performed an Hsc82 ATPase assay with increasing concentration of Ids2 or Ids2-S148D. The assay revealed that Ids2 triggers ATPase activity of Hsc82 and phosphorylation of Ids2 attenuates the reaction (*Figure 4A*). Hsc82 is required for mitochondrial F1-ATPase assembly (*Francis and Thorsness, 2011*). The *ids2Δ* and *ids2-S148D* strains displayed a growth defect in glycerol (*Figure 1C*), and formed white colonies (*Figure 4B*), which are caused by the loss of mitochondrial function in the W303-1A strain (*Wang et al., 2007*), implying that the phosphorylation of Ids2 may hamper the Hsc82 chaperone activity and thereby diminish the mitochondrial function.

We next asked whether Ids2 acts as a co-chaperone of HSP90 in the establishment of the HSP90-mediated protein-folding process. A firefly luciferase reporter assay was used to determine the ability of HSP90-dependent protein refolding (*Distel et al., 1992*). Western blot analysis revealed that the protein abundance of firefly luciferase was slightly reduced in *ids2Δ* cells compared to that in *hsc82Δ* cells at 30°C; however, luciferase expression was completely lost in *ids2-S148D* cells, while *ids2-S148A* cells expressed an amount of firefly luciferase equivalent to that of the wild-type (*Figure 4C*). Meanwhile, firefly luciferase could not be detected in the *ids2Δ* and *hsc82Δ* strains at 37°C. Since the accumulation of misfolded proteins would result in degradation (*Distel et al., 1992*), a mutant HSP90 chaperone or a dissociation of the chaperone folding network could cause mass accumulation and subsequent degradation of HSP90-dependent substrates. To further investigate the influence of phosphorylated Ids2 on the chaperone function of Hsc82, we conducted the glucocorticoid receptor (GR) maturation (*Schena and Yamamoto, 1988*) and v-Src toxicity (*Taipale et al., 2012*; *Nathan and Lindquist, 1995*) assays. GR activities in the *ids2Δ* and *ids2-S148D* strains decreased about 30% compared with that in the wild-type cells (*Figure 4—figure supplement 2A*). Consistently, the v-Src toxicity, which is strictly dependent on yeast HSP90 in vivo (*Taipale et al., 2012*; *Nathan and Lindquist, 1995*), reduced in the *ids2Δ* and *ids2-S148D* strains (*Figure 4—figure supplement 2B*). Taken together, these findings further elucidate that Ids2 may act as a co-regulator for the chaperone activity of HSP90.

Protein aggregates are found in a variety of diseases, including type II diabetes, Parkinson's disease, and Alzheimer's disease (*Chiti and Dobson, 2017*). In yeast, many cellular protein aggregates form during aging, for example, Gln1 aggregates with HSP90 upon glucose deprivation and cellular aging (*O'Connell et al., 2014*). Thus, we speculated that compromising the function of Ids2 may

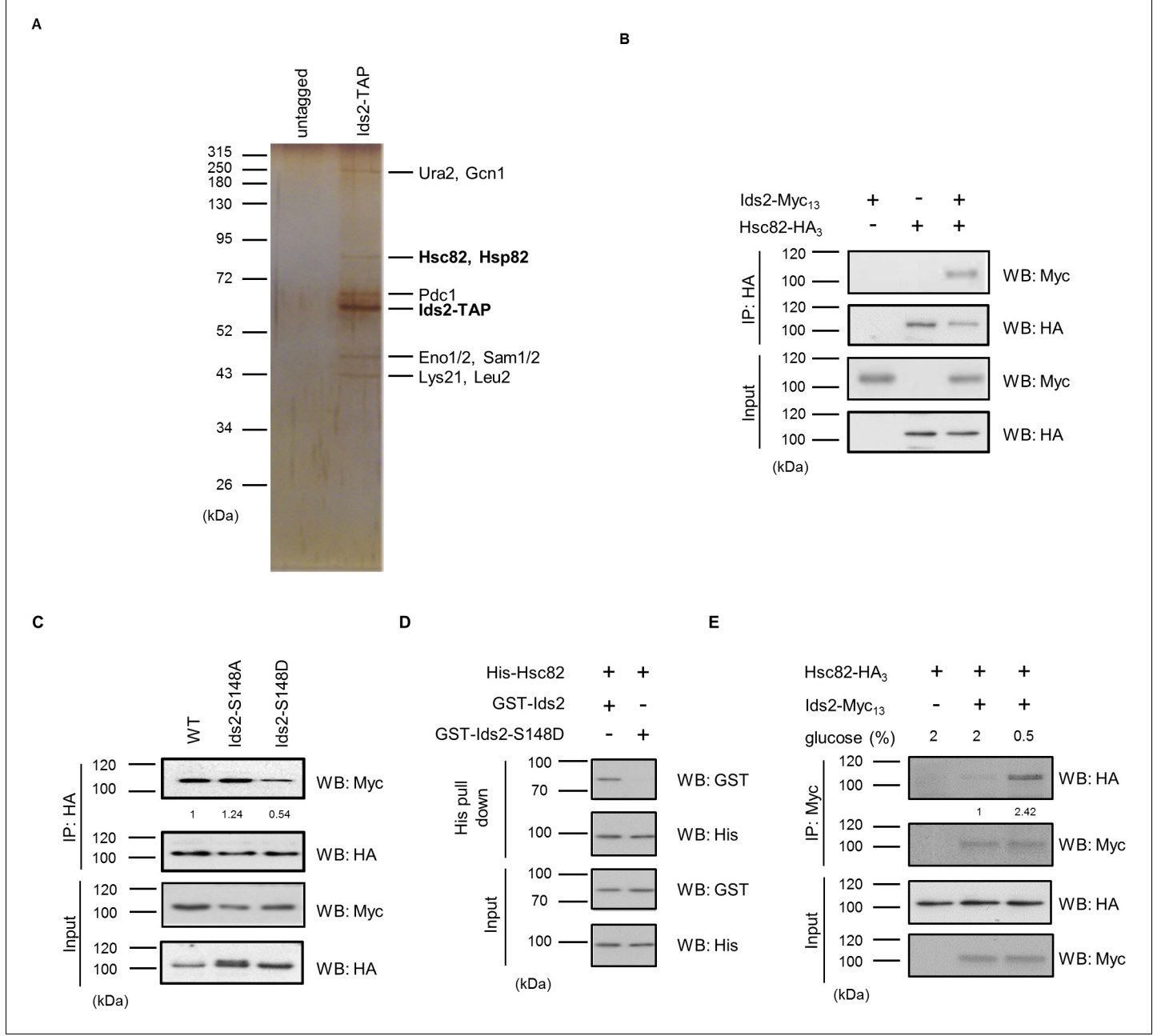

**Figure 3.** Ids2 interacts with HSP90 families.  (A) A silver-stained gel of the affinity-purified Ids2. Both HSP90 families, Hsc82 and Hsp82, were identified by LC-MS/MS analysis. (B) A co-immunoprecipitation assay was conducted using chromosomal-tagged Ids2-Myc$_{13}$ and Hsc82-HA$_3$. (C) Co-immunoprecipitation assay between Hsc82 and Ids2 mutants. The levels of signal compared with that of WT are shown below. (D) Purified recombinant GST-Ids2 and His$_6$-Hsc82 proteins were subjected to a His-tagged Metal Affinity Purification assay. (E) Tagged strains were incubated in 2% (normal) or 0.5% (*French et al., 2013*) glucose. Ids2-Myc$_{13}$ was used to co-immunoprecipitate Hsc82-HA$_3$. The numbers below are the means of the intensity ratios of HA/Myc compared with that of WT.

DOI: https://doi.org/10.7554/eLife.39925.009

The following source data and figure supplements are available for figure 3:

**Source data 1.** A list of peptides detected from Mass Spectrometry analysis of the Ids2-TAP co-purified proteins.
DOI: https://doi.org/10.7554/eLife.39925.012
**Figure supplement 1.** Ids2 formed a protein complex with HSP90.
DOI: https://doi.org/10.7554/eLife.39925.010
**Figure supplement 2.** Mapping the interacting domains between Ids2 and Hsc82 by the yeast two-hybrid assay.
DOI: https://doi.org/10.7554/eLife.39925.011

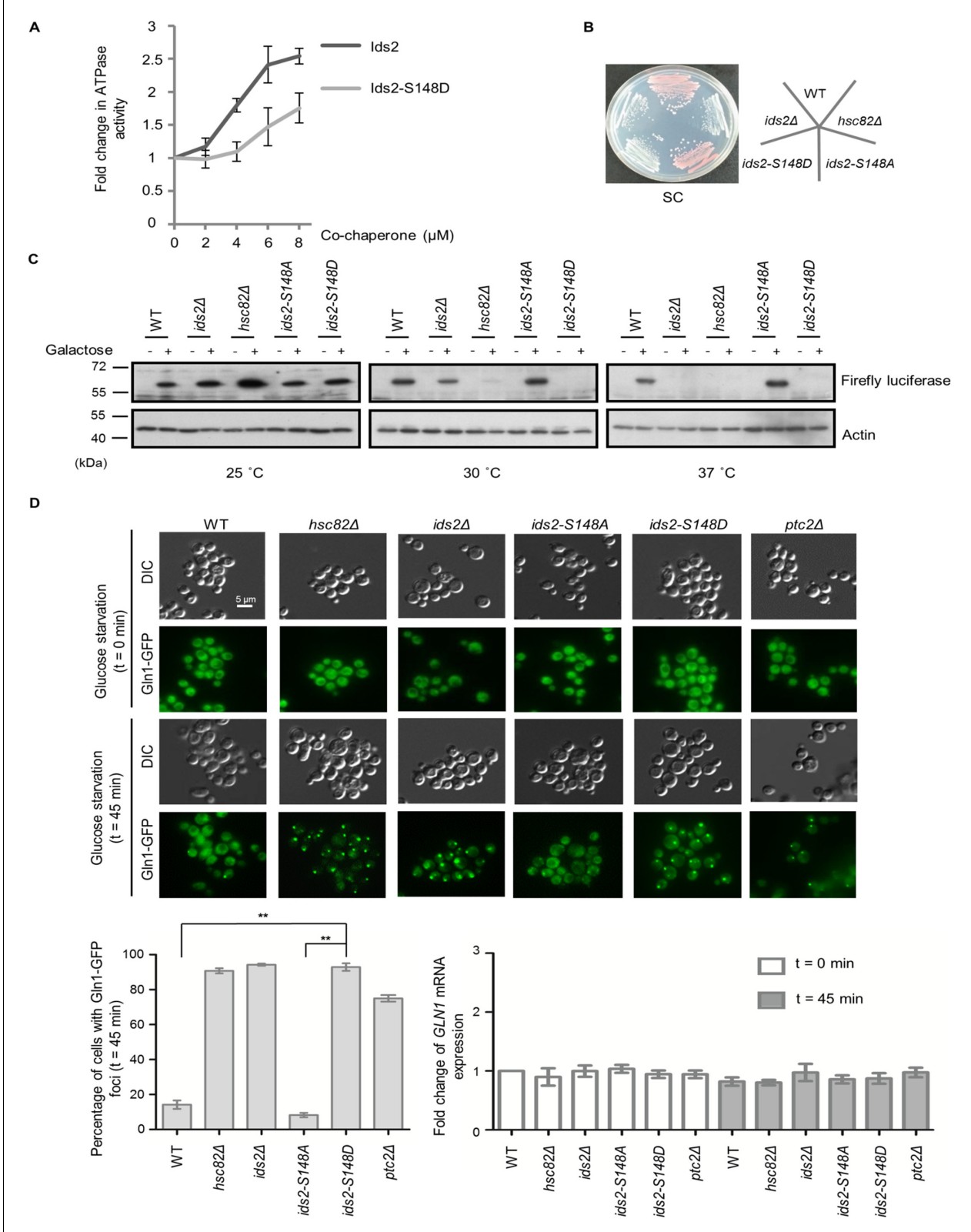

**Figure 4.** Ids2 participates in the HSP90-regulated protein-folding response. (A) The purified Hsc82 was mixed with Ids2 or Ids2-S148D, and the ATPase reaction was incubated for 90 min at 37°C. Error bars represent the standard deviation (SD) calculated from three independent experiments. (B) Cells, as indicated, were streaked on SC glucose, and the plate image was captured after 3 days. (C) An in vivo chaperone assay was conducted using an exogenously expressed firefly luciferase, and Western blot analysis was performed. (D) Endogenously expressed Gln1-GFP in WT and mutant cells

*Figure 4 continued on next page*

*Figure 4 continued*

was measured after 45 min without glucose. At least 200 cells were counted for each strain (**, p < 0.01, Student's t-test, two-tailed). The mRNA levels of *GLN1* were determined by quantitative RT-PCR relative to the housekeeping gene, *ACT1*. Error bars represent SD for three biological replicates.

DOI: https://doi.org/10.7554/eLife.39925.013

The following figure supplements are available for figure 4:

**Figure supplement 1.** The effect of heat-shock stress or *hsc82Δ* on the protein levels of Ids2.

DOI: https://doi.org/10.7554/eLife.39925.014

**Figure supplement 2.** The influences of phosphomimetic Ids2 on the chaperone function of Hsc82 and CLS.

DOI: https://doi.org/10.7554/eLife.39925.015

lead to aggregation of Gln1. We examined the Gln1-GFP localization in *ids2* phosphomimetic mutants under glucose deprivation (*Figure 4D*). The number of Gln1 foci in *ids2-S148D* mutants was dramatically increased compared to that in wild-type and *ids2-S148A* strains, implying that Ids2 phosphorylation may trigger Gln1 aggregation through down-regulation of HSP90 activity.

To determine if dephosphorylation of Ids2 is important for CR-induced lifespan extension, the CLS of *ids2Δ*, *hsc82Δ*, *ids2-S148A* and *ids2-S148D* were examined under CR (*Figure 4—figure supplement 2C*). CLS of the *ids2-S148A* strains under CR prolonged to the same extent as that of the wild-type strain, while the *ids2-S148D* strains lost CR-mediated lifespan extension. Therefore, dephosphorylated Ids2 is critical for the extended lifespan under CR.

## Discussion

Multiple approaches that integrate large genomic and proteomic datasets have facilitated the identification of factors governing longevity. However, crosstalk between these aging pathways is rare. Both nutrient sensing and proteostasis control lifespan. Here, our phosphoproteomic screen mapped 632 yeast proteins containing CR-responsive phosphorylation, although we could not rule out the possibility that some of the alterations might be due to a change in the abundance of the protein itself. Coupled with functional analyses, we identified a co-chaperone protein, Ids2, which is inactivated under glucose intake. The PKA writer and PP2C eraser may orchestrate glucose sensing and protein folding, and phosphorylation of Ids2 impedes its association with HSP90. Thus, glucose concentration may influence the HSP90 chaperone activity, enabling cells to control protein quality under environmental conditions for sustained longevity (*Figure 5*). Our findings are consistent with a previous large-scale analysis of the genetic interaction between HSP90 and Ids2 (*Franzosa et al., 2011*). We predict that Ids2 is a key regulator in promoting cellular tolerance to stress. HSP90 is a conserved protein controlling late protein-folding steps. Interestingly, sequence alignments of Ids2 show a certain degree of sequence similarity across homologues of HSP90 co-chaperones (*Figure 5—figure supplement 1*), including the S148 residue. It will be interesting to learn whether Ids2 homologues in other organisms also control lifespan.

Two lines of observation demonstrated that glucose intake modulates HSP90: high-glucose consumption boosts rat HSP90 expression (*Yang et al., 2016*) and PKA-mediated porcine HSP90α phosphorylation enhances the translocation of endothelial HSP90α to the cell surface (*Lei et al., 2007*). Through this study, we discovered a previously underappreciated glucose-mediated chaperone regulatory process: glucose concentration modulates HSP90 activity by PKA-dependent co-chaperone phosphorylation. Other than glucose consumption, heat shock (*Verna et al., 1997*) and oxidative stress (*Petkova et al., 2010*) have been shown to regulate the PKA pathway. Indeed, heat shock and oxidative stress both hamper Ids2 phosphorylation, suggesting that cells may use Ids2 dephosphorylation to stimulate chaperone activity for coping with various stresses.

We revealed that PP2C phosphatase antagonizes the function of PKA to extend lifespan. *S. cerevisiae* encodes seven phosphatases in the type 2C Ser/Thr phosphatase (PP2C) superfamily (Ptc1-Ptc7) (*Ariño et al., 2011*), and Ptc1, Ptc2, and Ptc3 are negative regulators of MAPK (*Warmka et al., 2001*). In deficiencies of the MAPK pathway, cells elevate their stress tolerance ability to extend CLS (*Aluru et al., 2017*). Although currently there is no direct evidence that PP2C may control stress tolerance through the MAPK pathway, our study provides a means by which PP2C governs stress tolerance: PP2C removes PKA-mediated Ids2 phosphorylation to maintain ample chaperone activity.

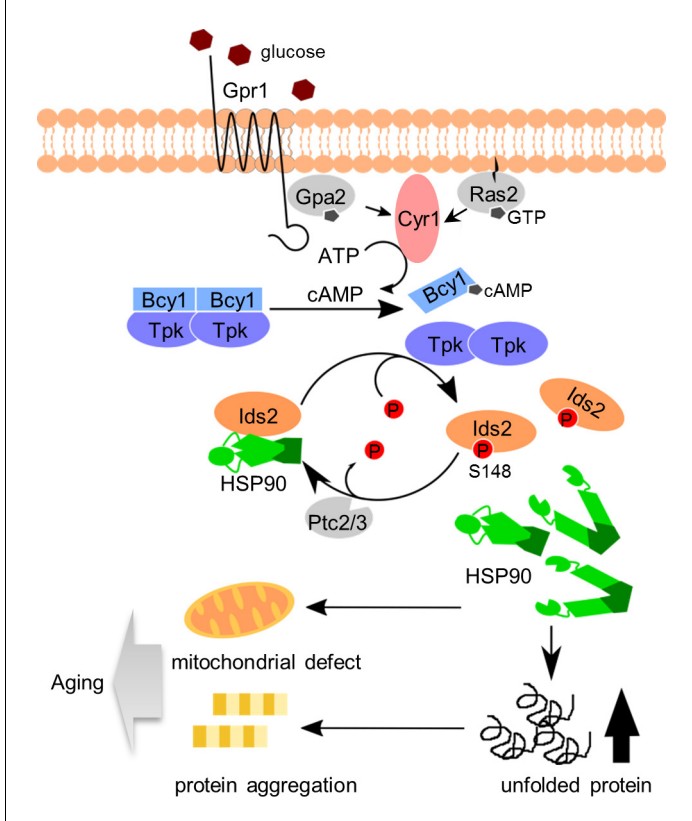

**Figure 5.** A proposed model to describe how glucose intake down-regulates chaperone activity and leads to aging. After glucose ingestion, cells convert ATP to cAMP to activate the PKA pathway. Activated PKA phosphorylates co-chaperone Ids2 to dissociate it from the HSP90 chaperone and reduce chaperone function, eventually leading to protein unfolding, mitochondrial dysfunction, and protein aggregation. However, under CR, dephosphorylated Ids2 might be essential for the full function of HSP90 to extend lifespan.

DOI: https://doi.org/10.7554/eLife.39925.016

The following figure supplement is available for figure 5:

**Figure supplement 1.** An alignment of Ids2 and its homologs.

DOI: https://doi.org/10.7554/eLife.39925.017

Finally, what is the role of Ids2 in protein folding at the molecular level? HSP90 forms a homodimeric complex, which consists of an N-terminal nucleotide-binding domain (NBD), a middle domain (M), and a C-terminal dimerization domain (CTD) (*Ali et al., 2006*). Co-chaperones can bind to specific domains of HSP90 to stabilize its conformation and further modulate its function. For example, yeast Sti1/Hop and human FKBP51 and FKBP52 utilize their tetratricopeptide (TPR) domains to interact with the C-terminal EEVD motif of HSP90 (*Röhl et al., 2013*) to stabilize chaperones. Moreover, the N-terminal region of co-chaperone Aha1 interacts with the M domain of HSP90 to stabilize its closed state conformation and thereby increase its ATPase activity (*Meyer et al., 2004*). Our domain mapping results determined that Ids2 also binds to the M domain of HSP90 and stimulates the ATPase activity of HSP90. In addition, Ids2 forms complex with Aha1 in our MS/MS data of the Ids2-TAP complex. Thus, it will be of interest to investigate the different controlling mechanisms and client specificities of Ids2 and Aha1 in modulating HSP90. Much work will be required to understand the mechanistic role of co-chaperone Ids2.

## Materials and methods

### Key resources table

| Reagent type (species) or resource | Designation | Source or reference | Identifiers | Additional information |
|---|---|---|---|---|
| Antibody | anti-Myc (mouse monoclonal) | Roche | 11667149001; RRID: AB_390912 | (1:2000) |
| Antibody | anti-HA (12CA5) (mouse monoclonal) | Roche | 11583816001; RRID: AB_514506 | (1:5000) |
| Antibody | anti-ACTIN (rabbit polyclonal) | Sigma-Aldrich | A2066; RRID:AB_476693 | (1:2000) |
| Antibody | anti-Hsp90 (rabbit polyclonal) | PMID: 23434373 | | (1:10000) |
| Antibody | anti-GST (rabbit polyclonal) | GeneTex | GTX110736; RRID: AB_1949427 | (1:2000) |
| Antibody | anti-Pgk1 (mouse monoclonal) | Invitrogen | 459250 | (1:5000) |
| Antibody | anti-(P)S148-Ids2 (rabbit purified polyclonal antibody) | This paper | | (1:1000) |
| Antibody | anti-Firefly Luciferase | GeneTex | GTX125849; RRID: AB_11173184 | (1:5000) |
| Commercial assay or kit | EnzChek phosphate assay kit | Thermo Fisher Scientific | E6646 | |
| Commercial assay or kit | KAPA SYBR FAST qPCR Master Mix Kit | Sigma-Aldrich | KK4600 | |
| Other | Calmodulin Sepharose 4B | GE Healthcare | 17-0529-01 | Affinity resin |
| Other | TALON Superflow | GE Healthcare | 28-9574-99 | Affinity resin |
| Other | IgG Sepharose 6 Fast Flow | Amersham Pharmacia Biotech AB | 17-0969-01 | Affinity resin |
| Other | Glutathione Sepharose 4 Fast Flow | GE Healthcare | 17-5132-01 | Affinity resin |
| Chemical compound, drug | Deoxycorticosterone acetate | Sigma-Aldrich | D7000 | |
| Chemical compound, drug | Protease inhibitor cocktail tablets | Roche | 4693132001 | |
| Chemical compound, drug | PKA, catalytic subunit, bovine heart | Millipore | 539576 | |
| Software | Image J | | RRID: SCR_003070 | Image analysis |

*Continued on next page*

*Continued*

| Reagent type (species) or resource | Designation | Source or reference | Identifiers | Additional information |
|---|---|---|---|---|
| Software | GraphPad Prism5 | | RRID: SCR_002798 | Statistical analysis and graph representation |

## Plasmids and yeast strains

The yeast strains and plasmids used in this study are listed in *Supplementary file 1*. Standard genetic and cloning methods were used for all constructions. pYES2-Ids2-TAP was generated by ligation of a PCR product from the Ids2-TAP HISMX6 strain into the KpnI-XhoI-treated pYES2. Kanamycin-resistant knockout strains were constructed through the transformation of kanamycin selection cassette fragments derived from PCR of BY4741. pRS306-*ids2* and other yeast integrating plasmids were PCR-amplified and cloned into pRS306. Single-point or multiple-site mutations were introduced into pRS306-based plasmids by site-directed mutagenesis. The oligonucleotides designed for mutagenesis and double crossover are listed in S4 Data. To generate chromosomal mutants of these genes, the pRS306-based plasmids were linearized by the appropriate restriction enzymes and transformed into the wild-type strain. After confirming the pop-in structure by Southern blot analysis and 5-FOA selection, the *URA3* pop-out mutants were selected and verified by PCR and sequencing. The C-terminal-tagged Gln1 was created by insertion of the PCR products of the GFP-*HIS3* cassette.

## Sample preparation for phosphopeptide enrichment

Yeast cells were grown in yeast extract peptone dextrose (YPD) medium (1% yeast extract, 2% bacto-peptone) supplemented with 2% (normal condition) or 0.5% glucose (calorie-restricted) for 3 hr and then harvested from log-phase growing cultures. Protein samples were extracted and subjected to gel-assisted digestion (*Wang et al., 2010*). First, proteins were fixed by sodium dodecyl sulfate-polyacrylamide gel electrophoresis (SDS-PAGE) directly, and the gel was cut into five pieces for trypsin proteolytic digestion. Digested peptides were extracted three times for 30 mins each and completely dried by vacuum centrifugation at room temperature. Then, we used a homemade immobilized metal affinity column for phosphopeptide enrichment (*Kanehisa and Goto, 2000*). An Ultra-Performance LC$^{TM}$ system (Waters) was used for automated purification of phosphopeptides. The Ni$^{2+}$ ions were removed, and the nitrilotriacetic acid resin was activated. The peptide samples from trypsin digestion were re-established in loading buffer and loaded into an activated immobilized metal affinity chromatography column. Finally, the unbound peptides were removed, and the bound peptides were eluted.

## LC-MS/MS Analysis

The purified phosphopeptides were analyzed in triplicate LC-MS/MS by an LTQ-Orbitrap XL hybrid mass spectrometer interfaced with an Agilent 1100 series HPLC. Survey full-scan MS spectra were acquired in the Orbitrap (*m/z* 350–1600) with the resolution set to 60,000 at *m/z* 400 and the automatic gain control target at 106. The 10 most intense ions were sequentially isolated for an MS/MS scan using collision-induced dissociation and detection in the linear ion trap with previously selected ions and dynamic exclusion for 90 s. All the measurements in the Orbitrap were performed with the lock mass option for internal calibration. Raw MS/MS data from the LTQ-Orbitrap were transformed to msm files using RAW2MSM software (version 1.1). The msm files were searched using Mascot (version 2.2.1) against the Swiss-Prot *Saccharomyces cerevisiae* database (version 54.2, 6493 sequences) with the following exceptions: only tryptic peptides with up to two missed cleavage sites were allowed, the fragment ion mass tolerance was set at 10 ppm, and the parent ion tolerance was set at 0.6 Da. Phosphorylation (*Portela et al., 2002*) and oxidation (M) were specified as variable modifications. Peptides were considered identified if their Mascot individual ion score was greater than 20 (*p*-0.05). The false discovery rates for Orbitrap data were determined with a Mascot score greater than 20 (*p*-0.05). The quantitative analysis of phosphopeptides was performed with the SEMI label-free algorithm (*Roskoski, 1983*). The raw data files acquired from the LTQ-Orbitrap were converted into files of the mzXML format by the program ReAdW, and the search results in MASCOT were

exported in Extensible Markup Language data (.xml) format. After data conversion, the confident peptide identification results ($p$-0.05) from each LC-MS/MS run were loaded and merged to establish a global peptide information list (sequence, elution time, and mass-to-charge ratio). Alignment of elution time was then performed based on the peptide information list using linear regression in different LC-MS/MS runs followed by correction of aberrational chromatographic shifts across fragmental elution-time domains. The mass spectrometry raw datasets were deposited to the ProteomeXchange Consortium (*Vizcaíno et al., 2014*; http://proteomecentral.proteomexchange.org) dataset identifier PXD001368 and DOI 10.6019/PXD001368. The quantitation results for phosphopeptides listed in S1 Data were obtained using IDEAL-Q software and further checked manually (*Ashburner et al., 2000*).

## Bioinformatics analysis - Gene Ontology enrichment analysis and KEGG orthology analysis

To functionally define the genes identified in this study, we performed Gene Ontology (GO), network, and pathway analyses using the PANTHER classification system (*Mi et al., 2013*; http://www.pantherdb.org/). The genes showing a 2-fold change with Bonferroni multiple correction testing were annotated for GO terms. The spatial-compartmental relationship and the genome annotation between the phosphopeptides were defined through the use of KEGG (Kyoto Encyclopedia of Genes and Genomes) orthology, a genomic, biochemical and enzyme-substrate metabolism-based online database developed by the team of Minoru Kanehisa of Kyoto University (*Dennis et al., 2003*).

## Stress resistance assay

Yeast cells were first grown overnight in yeast extract peptone (YEP) or synthetic complete (*Vizcaíno et al., 2014*) glucose medium at 30°C. Then, cells were re-inoculated into fresh YEP or SC medium and grown to exponential phase ($OD_{600}$ = 0.5). Tenfold serial dilutions of indicated strains were spotted onto YEP with 2% glucose (YEPD), YEP with 3% glycerol (YEPG), or SC glucose plates containing various concentrations of chemicals: 0.04% Methyl methanesulfonate (MMS), 100 mM hydroxyurea, 10 mM $H_2O_2$, 400 mM LiCl, 50 µg/ml hygromycin B (Hyg B), and 5 nM rapamycin, and incubated at 30°C (normal) or 37°C (heat shock) for 2–3 days.

## Chronological lifespan assay

The CLS assay of wild-type, *ids2Δ*, *ids2-S148A*, and *ids2-S148D* strains were carried out as previously described (*Postnikoff and Harkness, 2014*; *Longo et al., 2012*). Viability at day 3, when the yeast had reached stationary phase, was defined as the point of 100% survival. Thereafter, aliquots from the culture were diluted according to the estimated survival and plated on YEPD plates every other day. After 3 days of incubation at 30°C, colony-forming units were calculated. A semi-quantitative CLS spotting assay was also performed as previously described (*Smith et al., 2007*). In brief, the same cell numbers from the culture were serially diluted 10-fold and spotted on YEPD plates every 2 days.

## Tandem Affinity Purification

Tandem affinity purification of Ids2 was performed as previously described (*Tsai et al., 2008*) in the protocol from the Yeast Resource Center of the University of Washington. In brief, the BJ2168 pYES2-Ids2-TAP strain was inoculated into one liter SC with 2% raffinose and grown to log phase prior to adding galactose to induce protein overexpression. Cells were lysed with lysis buffer (150 mM NaCl, 1% Nonidet P-40, 50 mM Tris-HCl pH 7.5, and protease inhibitors) and with the use of glass beads and a homogenizer. Then, 250 µL of IgG Sepharose 6 Fast Flow prep in 1:1 slurry in NP-40 was added to the lysate and incubated at 4°C for 2 hr with gentle shaking. Beads were washed three times with lysis buffer and once with TEV cleavage buffer. Fifty units of TEV protease (Thermos) was added, and the solution was incubated overnight. The supernatant was collected, and calmodulin-binding buffer (25 mM Tris-HCl pH 8.0, 150 mM NaCl, 1 mM Mg acetate, 1 mM imidazole, 2 mM $CaCl_2$, 10 mM β-mercaptoethanol, 0.1 % NP-40) and 200 µL of calmodulin sepharose beads (GE Healthcare) were added for a 2 hr incubation at 4°C. Samples were washed, and the bound proteins were eluted into fractions with calmodulin elution buffer (25 mM Tris-HCl, pH 8.0, 150 mM NaCl, 1

mM Mg acetate, 1 mM imidazole, 20 mM EGTA, 10 mM β-mercaptoethanol, 0.1 % NP-40). The eluents were combined and precipitated with 25% trichloroacetic acid. The samples were analyzed with 10% SDS-PAGE gels and stained with silver stain. Bands were cut out and trypsinized for mass spectrometry analysis.

## One-dimensional SDS-PAGE combined nano-LC-MS/MS (GeLC-MS/MS)

To identify the Ids2-TAP interacting proteins, GeLC-MS/MS, one of the targeted proteomics approaches, was applied as described previously (*Liu et al., 2015*). Briefly, gel pieces were destained in 50 mM $NH_4HCO_3$/ACN (3:2, v/v) three times for 25 min each, dehydrated in ACN and dried in a SpeedVac. In-gel proteins were reduced with 10 mM dithiothreitol in 25 mM $NH_4HCO_3$ at 56°C for 45 min, allowed to stand at room temperature (RT) for 10 min, and then alkylated with 55 mM iodoacetamide in the dark for 30 min at RT. After the proteins were digested by sequencing grade modified porcine trypsin (1:100; Promega, Madison, WI) overnight at 37°C, peptides were extracted from the gel with ACN to a final concentration of 50%, dried in a SpeedVac, and then stored at 20°C for further use. For reverse-phase LC-MS/MS analysis, each peptide mixture was resuspended in HPLC buffer A (0.1% formic acid, Sigma, St. Louis, MO) and loaded into a trap column (Zorbax 300 SB-$_{C18}$, 0.3 × 5 mm, Agilent Technologies, Wilmington, DE) at a flow rate of 0.2 µL/min in HPLC buffer A. The salts were washed with buffer A at a flow rate of 20 µL/min for 10 min, and the desalted peptides were then separated on a 10 cm analytical $C_{18}$ column (inner diameter, 75 µm). The peptides were eluted by a linear gradient of 0–10% HPLC buffer B (99.9% ACN containing 0.1% formic acid) for 3 min, 10–30% buffer B for 35 min, 30–35% buffer B for 4 min, 35–50% buffer B for 1 min, 50–95% buffer B for 1 min, and 95% buffer B for 8 min at a flow rate of 0.25 µL/min across the analytical column. The LC setup was coupled online to an LTQ-Orbitrap linear ion trap mass spectrometer (Thermo Scientific, San Jose, CA) operated using Xcalibur 2.0.7 software (Thermo Scientific). Full-scan MS was performed using the Orbitrap in an MS range of 400–2000 Da, and the intact peptides were detected at a resolution of 30,000. Internal calibration was performed using the ion signal of cyclosiloxane peaks at m/z 536.165365 as a lock mass. A data-dependent procedure was applied that alternated between one MS survey scan and six MS/MS scans for the six most abundant precursor ions in the MS survey scan with a 2 Da window and fragmentation via CID with 35% normalized collision energy. The electrospray voltage applied was 1.8 kV. Both MS and MS/MS spectra were acquired using one microscan with a maximum fill-time of 1000 and 150 ms for MS analysis, respectively. Automatic gain control was used to prevent overfilling of the ion trap, and $5 \times 10^4$ ions were accumulated in the ion trap to generate the MS/MS spectra. All the MS and MS/MS data were analyzed and processed using Proteome Discoverer (version 1.4, Thermo Scientific). The top six fragment ions per 100 Da of each MS/MS spectrum were extracted for a protein database search using the Mascot search engine (version 2.4, Matrix Science) against the UniProtKB/Swiss-Prot sequence database. The top-six-peaks filter node improved the number of peptides identified with high confidence by reducing the number of peaks in the searched peak lists. This method avoids matching peptide candidates to spurious or noise peaks, thereby avoiding false peptide matches. The search parameters were set as follows: carbamidomethylation (C) as the fixed modification, oxidation (M), N-acetyl (protein), pyro-Glu/Gln (N-term Q), six ppm for MS tolerance, 0.8 Da for MS/MS tolerance, and two for missing cleavage. After database searching, the following filter criteria were applied to all the results: a minimum peptide length of six, a minimum of two unique peptides for the assigned protein, and peptide and protein identification with FDR less than 1% peptide filter were accepted. For precursor ion quantification, Proteome Discoverer was employed using a standard deviation of 2 ppm mass precision to create an extracted ion chromatogram (EIC) for the designated peptide.

## Antibody generation and western blot analysis

Cells were grown at 30°C in YEPD medium. Cell lysates were prepared with lysis buffer (150 mM NaCl, 1% nonidet P-40, 1% deoxycholate, 0.1% SDS, 50 mM Tris-HCl, pH 7.5, and protease inhibitors) or precipitated with trichloroacetic acid. To generate antibodies, rabbits were boosted with carrier-conjugated phosphopeptides once per month. Pre-immune sera were collected before boosting. Blood was collected every 2 weeks, incubated at 37°C for 30 mins, and separated via high-speed centrifugation. Clarified serum was incubated at 56°C for 30 mins to remove complement.

The specificity of antibodies was verified by means of peptide dot blot analysis. Images were captured and quantified by a bioluminescence imaging system (UVP BioSpectrum AC Imagine System, UVP).

## In Vitro kinase and phosphatase assays

The Ids2 ORF was cloned into the EcoRI and XhoI sites of pGEX-4T-1 for expression of GST-Ids2. A point mutation of Ids2-S148 was created to express GST-Ids2-S148A. Bovine heart catalytic subunit C ($C_b$) was purchased from Millipore and PKI 6–22 (Millipore) was used as a negative control. PKA was obtained from crude extracts of yeast strain KT1115, and the PKA kinase-dead strain was used as a negative control (*Toone and Jones, 1998*). For the phosphatase assay, the full-length Ptc2 and the C terminal-truncated Ptc2 (1-312) sequences were cloned into the BamHI and SalI sites of pGEX-4T-1 for the expression of GST-Ptc2 and GST-Ptc2 (1-312)-kinase-dead, respectively. GST proteins were expressed in the BL21 (DE3) pLysS strain in LB plus Amp and induced with 1 mM isopropyl-1-thio-β-D-galactopyranoside (IPTG) at 37°C for 3 hr. The GST-tagged proteins were purified with Glutathione Sepharose 4B (GE) in binding buffer (140 mM NaCl, 2.7 mM KCl, 10 mM $Na_2HPO4$, 1.8 mM $KH_2PO4$ pH 7.5, 1% Triton X-100, 10% glycerol) for 1 hr and eluted with elution buffer (50 mM Tris-HCl, 10 mM reduced glutathione pH 8.0).

The kinase assay was started by mixing different sources of PKA (50 mM extracts from KT1115) with GST-Ids2 substrates in kinase buffer (50 mM potassium phosphate pH 7.5, 0.1 mM EGTA, 0.1 mM EDTA, 15 mM $MgCl_2$, 10 mM β-mercaptoethanol, 100 mM ATP). After 15 min at 30°C, the samples were analyzed by Western blotting with Ids2 S148 phosphospecific antibodies. For the isotope assay, additional γ-$^{32}$P-ATP was added to the mixture, and the aliquots were processed according to the phosphocellulose paper method (*Toone and Jones, 1998*).

For the phosphatase assay, phosphorylated Ids2 was generated by incubating GST-Ids2 with bovine heart catalytic subunit C ($C_b$) in kinase buffer for 15 min at 30°C. The reaction was stopped by incubating with PKI 6–22 at 30°C for 10 min. GST-Ptc2 or GST-Ptc2 (1-312) was then added to the mixture for the phosphatase reaction. After 15 min at 30°C, samples were analyzed by Western blotting with Ids2 S148 phosphospecific antibodies.

## Co-immunoprecipitation assay

W303 strains containing chromosomally-tagged *HSC82*-HA$_3$ and *IDS2* plasmids (pRS426-*IDS2*-Myc$_{13}$, pRS426-*ids2-S148A*-Myc$_{13}$, or pRS426-*ids2-S148D*-Myc$_{13}$) were grown to exponential phase, and cells were harvested. Pellets were resuspended in lysis buffer (50 mM NaCl, 0.1 % NP-40, 150 mM Tris-HCl pH 8.0) supplemented with protease inhibitors (Roche). Cells were broken by a *Fast-Prep*-24 5G Homogenizer (MP biomedical) and supernatants were collected after centrifugation. The supernatants were mixed with either anti-HA (Roche) or anti-Myc (Roche) antibodies, and followed by incubation with protein G Sepharose beads. Immunoprecipitates were washed four times with lysis buffer and then eluted by boiling in sample buffer. Samples were resolved by 7% SDS-PAGE and analyzed by Western blotting using the appropriate antibodies.

## Yeast Two-hybrid analysis

To investigate the interaction between Ids2 and Hsc82, a DNA fragment encoding the full length or N-terminal amino acids of Ids2 was ligated downstream of the Gal4 DNA-binding domain in pGBDU-C1. A fragment encoding the full length or N-terminal amino acids of Hsc82 was inserted in pGAD-C1 downstream of the Gal4 DNA-activating domain. The resulting plasmids were transformed into PJ69-4A and Leu$^+$ and Ura$^+$ cells were selected. Cells were restreaked on SC-Leu-Ure-Ade-His glucose plates to inspect the bait-prey reciprocity.

## His-tagged metal affinity purification assay

An expression construct bearing Hsc82 with an N-terminal His$_6$-tag was transformed into *E. coli*. *E. coli* cultures were grown in LB media to an $OD_{600}$ of 0.8 and then induced with 0.5 mM IPTG for 4 hr. The His$_6$-tag proteins were purified with TALON His-tagged Protein Purification Resins (GE) in binding buffer (50 mM Tris-HCl pH 8.0, 100 mM NaCl, 50 mM imidazole, 1% Triton X-100) for 1 hr and eluted with elution buffer (50 mM Tris-HCl pH 8.0, 250 mM imidazole, 10% glycerol).

For the His$_6$-Hsc82 pull-down assay, purified His$_6$-Hsc82 were incubated with purified GST-Ids2 or GST-Ids2-S148D in incubation buffer (20 mM Tris-HCl pH 7.5, 50 mM NaCl, 5% glycerol, 0.01% Triton X-100, 2 mM DTT) at 4°C for 1 hr. The reaction was precipitated with TALON His-tagged Protein Purification Resins (GE) at 4°C for 1 hr. After washing three times with incubation buffer, the His$_6$-tagged Hsc82 and co-purified Ids2 proteins were recovered by elution with imidazole.

## ATPase assay

For the ATPase assay, His$_6$-Hsc82, GST-Ids2, and GST-Ids2-S148D proteins were isolated on appropriate columns (a HisTrap FF column for His-tagged proteins and a GSTrap FF column for GST-tagged proteins) using AKTA Explorer FPLC (GE Healthcare) and further purified by size exclusion chromatography on a Superdex 200 column (GE healthcare) in 25 mM Hepes pH 7.2, 50 mM NaCl, and 5 mM β-mercaptoethanol.

The ATPase activity of Hsc82 was measured by using EnzChek phosphate assay kit (Molecular Probes, Inc.) as previously described (*Wolmarans et al., 2016*). The His$_6$-Hsc82 protein (2 µM) was incubated with Ids2 or Ids2-S148D protein at 37°C with 1 mM ATPase buffer (40 mM HEPES pH 7.5, 150 mM KCl, 5 mM MgCl$_2$, 2 mM DTT). The reactions were incubated for 90 min and then diluted 1:5 into a 100 µl phosphate assay reaction as outlined by the manufacturer's instructions. A standard curve was generated using the inorganic phosphate solution provided by the manufacturer (Molecular Probes, Inc.).

## In Vivo chaperone assay

For the luciferase assay, a pRS316 plasmid containing the *GAL1-10* promoter and the SEL-allele of firefly luciferase (*Distel et al., 1992*) was purchased from Addgene and transformed into wild-type, *ids2Δ, hsc82Δ, ids2-S148A,* and *ids2-S148D* cells. Cells were inoculated into raffinose medium and cultured overnight prior to 3 hr galactose induction in 25°C temperatures. Then, the culture aliquoted and transferred to an incubator at 30°C and 37°C temperature, respectively. After 2 hr incubation, the protein was precipitated by TCA and Western blot analysis was performed with anti-firefly luciferase antibodies (Genetex).

The GR maturation assay was performed as previously described (*Zuehlke et al., 2017*). In brief, pG/N795 (Addgene) encoding mammalian steroid receptor GR and pUCΔSS-26X (Addgene) encoding the β-galactosidase reporter were transformed into wild-type, *ids2Δ, hsc82Δ, ids2-S148A,* and *ids2-S148D* cells. Each strain from the overnight culture was diluted to 0.4 OD$_{600}$ in fresh media containing 10 µM synthetic hormone analog deoxycorticosterone (Sigma). After 5 hr incubation, the cells were harvested and the β-galactosidase activity was measured.

For the v-Src toxicity assay, cells were transformed with plasmids containing either v-Src or c-Src in pY413 (*Boczek et al., 2015*). After stationary phase growth in glucose-containing selection medium, 3 µl aliquots from the serial dilutions were spotted on plates containing galactose and raffinose.

## Induction of Gln1 foci by glucose deprivation

Cells with Gln1-GFP expression were inoculated into fresh SC glucose selection medium to 0.3 OD/ml from an overnight culture. When cells were regrown to 1 OD/ml, cells were washed with PBS and then resuspended in SC medium without glucose for 45 min to induce foci formation. After induction, cells were harvested and fixed with 3.7% formaldehyde. Statistical significance was analyzed by the student's t-test using GraphPad Prism 5 (GraphPad Software, Inc.). The amount of *GLN1* and *ACT1* mRNA was quantified by real-time reverse transcription polymerase chain reaction (RT-PCR) (Kappa), using primers listed in *Supplementary file 1*.

## Acknowledgments

We thank Professors Silvia Rossi and Johannes Buchner for yeast strains and plasmids. We also thank Chia-Feng Tsai for his assistance of mass spectrometry analysis. This work was financially supported by the 'Center of Precision Medicine' from The Featured Areas Research Center Program within the framework of the Higher Education Sprout Project by the Ministry of Education (MOE) and the Ministry of Science and Technology (MOST 106–2311-B-002–010-MY3) in Taiwan.

## Additional information

### Funding

| Funder | Grant reference number | Author |
|--------|------------------------|--------|
| Ministry of Science and Technology, Taiwan | MOST106-2311-B-002-010-MY3 | Shu-Chun Teng |

The funders had no role in study design, data collection and interpretation, or the decision to submit the work for publication.

### Author contributions

Yu-Chen Chen, Data curation, Formal analysis, Validation, Investigation, Visualization, Methodology, Writing—original draft, Project administration, Analysis and interpretation of data; Pei-Heng Jiang, Formal analysis, Investigation, Visualization, Methodology, Project administration, Analysis and interpretation of data; Hsuan-Ming Chen, Chang-Han Chen, Analysis and interpretation of data; Yi-Ting Wang, Resources, Analysis and interpretation of data; Yu-Ju Chen, Chia-Jung Yu, Resources, Methodology, Writing—review and editing; Shu-Chun Teng, Conceptualization, Supervision, Funding acquisition, Validation, Methodology, Project administration, Writing—review and editing

### Author ORCIDs

Yu-Chen Chen (iD) http://orcid.org/0000-0001-8434-7084
Shu-Chun Teng (iD) http://orcid.org/0000-0002-6492-2560

### Decision letter and Author response

Decision letter https://doi.org/10.7554/eLife.39925.023
Author response https://doi.org/10.7554/eLife.39925.024

## Additional files

### Supplementary files

• Supplementary file 1. Yeast strains, plasmids, and primer sets used in this study.
DOI: https://doi.org/10.7554/eLife.39925.018

• Transparent reporting form
DOI: https://doi.org/10.7554/eLife.39925.019

### Data availability

The mass spectrometry proteomics data have been deposited to the ProteomeXchange Consortium via the PRIDE partner repository with the dataset identifier PXD001368 and DOI 10.6019/PXD001368.

The following dataset was generated:

| Author(s) | Year | Dataset title | Dataset URL | Database and Identifier |
|-----------|------|---------------|-------------|--------------------------|
| Chen Y-C, Jiang P-H, Chen H-M, Chen C-H, Wang Y-T, Chen Y-J, Yu C-J, Teng S-C | 2018 | Mass spectrometry proteomics data from: Glucose intake hampers PKA-regulated HSP90 chaperone activity | https://doi.org/10.6019/pxd001368 | ProteomeXchange, PXD001368 |

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
