## [Decision Letter]

Thank you for submitting your article "Glucose intake hampers PKA-regulated HSP90 chaperone activity" for consideration by *eLife*. Your article has been reviewed by three peer reviewers, and the evaluation has been overseen by a Reviewing Editor and Philip Cole as the Senior Editor. The reviewers have opted to remain anonymous.

The reviewers have discussed the reviews with one another and the Reviewing Editor has drafted this decision to help you prepare a revised submission.

Overall, the data in this paper position Ids2 as a previously unknown target of a phosphorylation switch and as a modulator of cellular stress, lifespan, and potentially as a regulator of Hsp90. These are important findings. However, the authors claim that these data highlight a previously unknown link between CR/lifespan and proteostasis is not supported, and this conclusion needs to be softened. In addition, direct effects of this putative co-chaperone on Hsp90 function, using established cell-based and in vitro assays, is essential to support the role of this factor on Hsp90 activity. In turn, the fact that Ids2 was reported as a DNA binding factor suggests instead that it might act with Hsp90 via the established role of Hsp90 and select co-chaperones as a regulator of transcription. This might explain changes in luciferase expression and Gln1-GFP aggregates presented and should be discussed.

The authors should make clear one major caveat of the initial phosphoproteomics data – that it is not normalized to changes in total protein levels upon caloric restriction. Therefore, it is not possible to differentiate between the phospopeptides that are changed due to a change in the abundance of the protein itself from those that are actually changed due to phosphorylation/desphosphorylation events.

The control for the Ids2-TAP IP-MS experiment is not ideal. A better control would be cells expressing just the TAP tag alone (Protein A + CBP), to discount non-specific hits that bind either of these two proteins. This is especially important given that Hsc82/Hsp82 are common contaminants in affinity proteomics experiments (as defined by the CRAPome database).

Finally, the authors suggest that this Ids2-dependent regulation of Hsc90 activity is key to longevity control, allows survival during chronological aging (starvation in stationary phase), and is required for CR to retard aging in stationary phase (it is stated, for example, that "PP2C phosphatase antagonizes the function of PKA to extend lifespan"). In this respect, the data does not at present support these conclusions and more data is required for making this link. For example, the authors should test whether Ids2 and Hsc90 are required for CR to prolong lifespan – no CR experiments are included presently. Also, for the conclusions stated it is expected that the *ids2-S148A* mutant should display a longer lifespan even without CR – is this true and if so, does it depend on Hsc90?

---

## [Author Response]

Overall, the data in this paper position Ids2 as a previously unknown target of a phosphorylation switch and as a modulator of cellular stress, lifespan, and potentially as a regulator of Hsp90. These are important findings. However, the authors claim that these data highlight a previously unknown link between CR/lifespan and proteostasis is not supported, and this conclusion needs to be softened. In addition, direct effects of this putative co-chaperone on Hsp90 function, using established cell-based and in vitro assays, is essential to support the role of this factor on Hsp90 activity. In turn, the fact that Ids2 was reported as a DNA binding factor suggests instead that it might act with Hsp90 via the established role of Hsp90 and select co-chaperones as a regulator of transcription. This might explain changes in luciferase expression and Gln1-GFP aggregates presented and should be discussed.

As suggested by the reviewers, we have softened our statements in Abstract, in Figure 5 legend, and in other places. To investigate the putative co-chaperone function on HSP90, the glucocorticoid receptor (GR) activity (Schena and Yamamoto, 1988) and v-Src toxicity (Boczek et al., 2015) assays have been performed. For monitoring GR maturation, the strains were co-transformed with a plasmid that constitutively expresses GR and a plasmid carrying a β-galactosidase reporter. We observed that the GR activity in the *ids2Δ* and its phosphomimetic mutant, *ids2-S148D,* decreased compared with that in the wild-type cells (new Figure 4—figure supplement 2A). Consistently, the v-Src toxicity, which is strictly dependent on yeast HSP90 in vivo, reduced in the *ids2D* and *ids2-S148D* strains (new Figure 4—figure supplement 2B). The results support our model that Ids2 may serve as a co-chaperone orchestrating the HSP90 function.

Although a potential DNA binding ability of Ids2 might have a chance to affect our luciferase expression and the Gln1-GFP aggregation results through transcriptional activation, in our firefly luciferase experiment, we expressed firefly luciferase under a *GAL1* promoter. The control samples were first grown at 25 °C and then shifted to higher temperatures (30 °C and 37 °C). There was no apparent decrease in firefly luciferase protein levels in *ids2Δ, hsc82Δ*, and *ids2-S148D* cells compared to that in the wild-type strain when we induced the expression with galactose at 25 °C (Figure 4C). Therefore, the reduction of the firefly luciferase protein under the higher temperature is probably not through an Ids2-mediated transcriptional activation on the *GAL1* promoter, but most likely due to protein degradation of misfolded firefly luciferase as Professor Susan Lindquist previously suggested (Nathan et al., 1997). Moreover, in the Gln1-GFP aggregation experiment, we have further examined the *GLN1* mRNA expression in the tested strains under glucose starvation using real-time quantitative reverse transcription PCR (new Figure 4D). The transcriptional level of *GLN1* did not show any significant change in these strains, indicating that the appearance of Gln1-GFP aggregates was not a consequence of the difference of transcriptional regulation of Ids2 on *GLN1*.

The authors should make clear one major caveat of the initial phosphoproteomics data – that it is not normalized to changes in total protein levels upon caloric restriction. Therefore, it is not possible to differentiate between the phospopeptides that are changed due to a change in the abundance of the protein itself from those that are actually changed due to phosphorylation/desphosphorylation events.

For the phosphoproteomic analysis used in this study, indeed we cannot rule out the possibility that the phosphopeptides were changed due to a change in the abundance of the protein itself. As suggested by the reviewer, we have added this possibility in Discussion (first paragraph). Moreover, to minimize this vagueness, we have marked the proteins (n = 16) in red in our mass data (new Figure 1—source data 1) that have been reported to be influenced on the transcriptional levels under CR stress according to a previous research (Lee and Lee, 2008). For the only positive candidate Ids2, we have verified that the protein level of Ids2 is not a major issue of regulation under CR (new Figure 2—figure supplement 1B).

The control for the Ids2-TAP IP-MS experiment is not ideal. A better control would be cells expressing just the TAP tag alone (Protein A + CBP), to discount non-specific hits that bind either of these two proteins. This is especially important given that Hsc82/Hsp82 are common contaminants in affinity proteomics experiments (as defined by the CRAPome database).

As suggested by the reviewer, we have added the cells expressing the TAP tag alone as a control (new Figure 3—figure supplement 1). We have used an HSP90 specific antibody to confirm that the signal was faithfully derived from HSP90 (Hsc82/Hsp82) protein and only appeared in the Ids2-TAP-pulldown, but not in the TAP-alone pulldown. The results indicate that the signal around 80 kDa on the silver staining gel may mainly come from the HSP90 proteins.

Finally, the authors suggest that this Ids2-dependent regulation of Hsc90 activity is key to longevity control, allows survival during chronological aging (starvation in stationary phase), and is required for CR to retard aging in stationary phase (it is stated, for example, that "PP2C phosphatase antagonizes the function of PKA to extend lifespan"). In this respect, the data does not at present support these conclusions and more data is required for making this link. For example, the authors should test whether Ids2 and Hsc90 are required for CR to prolong lifespan – no CR experiments are included presently. Also, for the conclusions stated it is expected that the ids2-S148A mutant should display a longer lifespan even without CR – is this true and if so, does it depend on Hsc90?

To test whether Ids2 and Hsc82 are required for CR-mediated lifespan extension, we have measured the chronological lifespan of *ids2Δ, hsc82Δ*, *ids2-S148A* and *ids2-S148D* under CR (0.5% glucose). Indeed, *ids2Δ, hsc82Δ*, and *ids2-S148D* strains lost their ability to extend lifespan under CR, and the *ids2-S148A* mutant showed lifespan extension as the wild-type control under CR (new Figure 4—figure supplement 2C). Moreover, the *ids2-S148A* mutant did not display a longer lifespan than the wild-type strain under normal condition. Based on these findings, we speculate that HSP90 needs dephosphorylated Ids2 for its full chaperone complex activity to extend lifespan under normal and CR conditions, and in yeast Ids2 may provide enough dephosphorylated form under normal condition for HSP90’s activity. We have softened our model (Introduction, first paragraph and Figure 5 legend, and other places) based on these observations.

References

Boczek EE, Reefschlager LG, Dehling M, Struller TJ, Hausler E, Seidl A, et al. Conformational processing of oncogenic v-Src kinase by the molecular chaperone Hsp90. *Proc Natl Acad Sci* U S A. 2015;112(25):E3189-98.

Lee YL, Lee CK. Transcriptional response according to strength of calorie restriction in *Saccharomyces cerevisiae. Mol Cells.* 2008;26(3):299-307.

Nathan DF, Vos MH, Lindquist S. in vivo functions of the *Saccharomyces cerevisiae* Hsp90 chaperone. *Proc Natl Acad Sci* U S A. 1997;94(24):12949-56.

Schena M, Yamamoto KR. Mammalian glucocorticoid receptor derivatives enhance transcription in yeast. *Science*. 1988;241(4868):965-7.